# Sequencing-based fine-mapping and in silico functional characterization of the 10q24.32 arsenic metabolism efficiency locus across multiple arsenic-exposed populations

Meytal Batya Chernoff[1,2,3]*, Dayana Delgado[1], Lin Tong[1], Lin Chen[1], Meritxell Oliva[1], Lizeth I. Tamayo[1], Lyle G. Best[4], Shelley Cole[5], Farzana Jasmine[1], Muhammad G. Kibriya[1], Heather Nelson[6], Lei Huang[7], Karin Haack[5], Jack Kent[5], Jason G. Umans[8,9], Joseph Graziano[5,10], Ana Navas-Acien[11], Margaret R. Karagas[12], Habib Ahsan[1,13,14], Brandon L. Pierce[1,13,15]*

1 Department of Public Health Sciences, University of Chicago, Chicago, Illinois, United States of America, 2 Interdisciplinary Scientist Training Program, University of Chicago, Chicago, Illinois, United States of America, 3 University of Chicago Pritzker School of Medicine, Chicago, Illinois, United States of America, 4 Missouri Breaks Industries Research Inc, Eagle Butte, South Dakota, United States of America, 5 Texas Biomedical Research Institute, San Antonio, Texas, United States of America, 6 School of Public Health, University of Minnesota, Minneapolis, Minnesota, United States of America, 7 Center for Research Informatics, University of Chicago, Chicago, Illinois, United States of America, 8 MedStar Health Research Institute, Hyattsville, Maryland, United States of America, 9 Georgetown-Howard Universities Center for Clinical and Translational Science, Georgetown University, Washington, District of Columbia, United States of America, 10 Department of Pharmacology, Columbia University, New York City, New York, United States of America, 11 Mailman School of Public Health, Columbia University, New York City, New York, United States of America, 12 Department of Epidemiology, Geisel School of Medicine at Dartmouth, Hanover, New Hampshire, United States of America, 13 Comprehensive Cancer Center, University of Chicago, Chicago, Illinois, United States of America, 14 Department of Medicine, University of Chicago, Chicago, Illinois, United States of America, 15 Department of Human Genetics, University of Chicago, Chicago, Illinois, United States of America

* mchernoff@uchicago.edu (MBC); bpierce@health.bsd.uchicago.edu (BLP)

**Data Availability Statement:** All summary statistics generated with this study are included

## Abstract

Inorganic arsenic is highly toxic and carcinogenic to humans. Exposed individuals vary in their ability to metabolize arsenic, and variability in arsenic metabolism efficiency (AME) is associated with risks of arsenic-related toxicities. Inherited genetic variation in the 10q24.32 region, near the arsenic methyltransferase (*AS3MT*) gene, is associated with urine-based measures of AME in multiple arsenic-exposed populations. To identify potential causal variants in this region, we applied fine mapping approaches to targeted sequencing data generated for exposed individuals from Bangladeshi, American Indian, and European American populations (n = 2,357, 557, and 648 respectively). We identified three independent association signals for Bangladeshis, two for American Indians, and one for European Americans. The size of the confidence sets for each signal varied from 4 to 85 variants. There was one signal shared across all three populations, represented by the same SNP in American Indians and European Americans (rs191177668) and in strong linkage disequilibrium (LD) with a lead SNP in Bangladesh (rs145537350). Beyond this shared signal, differences in LD patterns, minor allele frequency (MAF) (e.g., rs12573221 ~13% in Bangladesh ~0.2% among American Indians), and/or heterogeneity in effect sizes across populations likely contributed

within the manuscript and its supporting information files. Individual-level data requests for all the data underlying results presented in the study can be requested by contacting IPPH@uchicago.edu. Requests will then be routed to the three individual studies, as there are different mechanisms for data access for each study. Normalized expression matrices, summary statistics for eQTLs and mQTLs, and covariates used for QTL mapping are available at the GTEx Portal (https://gtexportal.org/home/datasets). DNAm normalized data is available at GEO (GSE213478); access to the DNAm raw data is provided through the AnVIL platform (https://anvil.terra.bio/#workspaces/anvil-datastorage/AnVIL_GTEx_V9_hg38). All GTEx protected data are available via dbGaP (phs000424.v9 and phs000424.v8.p2). GTEx whole genome sequencing data can be requested through dbGaP (https://gtexportal.org/home/protectedDataAccess).

**Funding:** Targeted DNA sequencing across all cohorts was supported by National Institutes of Health grant R01 ES023834 (to B.L.P.) The Health Effects of Arsenic Longitudinal Study was supported by National Institutes of Health grants R35 ES028379 (to B.L.P.), R21 ES024834 (to B.L.P.), P42ES010349 (to J.G.), R01 CA107431 (to H.A.), P30 ES027792 (to H.A.), R24 ES028532 (to H.A.), and R24 TW009555 (to H.A.). The New Hampshire Skin Cancer Study was supported by U.S. National Institutes of General Medicine grant P20GM104416 (to M.R.K.) and by National Institutes of Health grants P42ES007373 (to C.Y. Chen) and R01CA057494 (to M.R.K.). The Strong Heart Study has been funded in whole or in part with federal funds from the National Heart, Lung, and Blood Institute, National Institute of Health, Department of Health and Human Services, under contract numbers 75N92019D00027, 75N92019D00028, 75N92019D00029, & 75N92019D00030. The study was previously supported by research grants: R01HL109315, R01HL109301, R01HL109284, R01HL109282, and R01HL109319 and R01HL090863, and by cooperative agreements: U01HL41642, U01HL41652, U01HL41654, U01HL65520, and U01HL65521 and from the National Institute of Environmental Health Sciences (P42ES033719, R01ES032638 and past grant R01ES021367). The content is solely the responsibility of the authors and does not necessarily represent the official views of the National Institutes of Health. This work was also supported by the National Institute of Environmental Health Sciences under the award number R35ES028379-03S1 (B.L.P.), the National Institute of General Medicine under award number

to the apparent population specificity of the additional identified signals. One of our potential causal variants influences *AS3MT* expression and nearby DNA methylation in numerous GTEx tissue types (with rs4919690 as a likely causal variant). Several SNPs in our confidence sets overlap transcription factor binding sites and cis-regulatory elements (from ENCODE). Taken together, our analyses reveal multiple potential causal variants in the 10q24.32 region influencing AME, including a variant shared across populations, and elucidate potential biological mechanisms underlying the impact of genetic variation on AME.

## Author summary

Inorganic arsenic is highly toxic, and exposure to arsenic increases risk for multiple diseases, including cancer. Individuals differ in their ability to metabolize and excrete arsenic, in part due to inherited genetic variation in and around the *AS3MT* gene, and these differences impact arsenic toxicity risk. To identify candidate causal variants in the *AS3MT* region, we applied fine-mapping methods to targeted sequencing data from The Health Effects of Arsenic Longitudinal Study (HEALS), the Strong Heart Study (SHS), and the New Hampshire Skin Cancer Study (NHSCS) (Bangladesh, American Indian, and European American populations). We detected 3 independent association signals in HEALS, 2 in SHS, and 1 in NHSCS; and we identified a set of candidate causal variants for each of these signals. One of the identified signals represents a potential causal variant that impacts arsenic metabolism across all three populations. Using omics-QTL co-localization analyses, we show that some of the variants identified act through regulation *AS3MT* in multiple tissue types. Overall, this work increases our understanding of variation in the *AS3MT* region and its role in arsenic metabolism across populations.

## Introduction

Arsenic-contaminated groundwater is a global public health issue, impacting >220 million individuals worldwide, with >85% of highly exposed individuals living in South Asia based on an exposure level of $\geq$10μg/L in drinking water [1,2]. The International Agency for Research on Cancer (IARC) classifies inorganic arsenic (iAs) as a "Group 1" human carcinogen [3] with chronic exposure increasing the risk of bladder [4], kidney, lung [5], liver, and skin cancers [3,6]. Exposure to iAs is also associated with increased risk for diabetes [7], as well as cardiovascular, cerebrovascular, and neurologic diseases [8–11]. A hallmark of chronic iAs exposure is the appearance of skin lesions, arsenical hyperkeratosis, typically on the hands and feet of exposed individuals [12,13]. The most common source of iAs exposure is contaminated drinking water [14] with ~2.1 million individuals in the U.S. [15] and 35 to 77 million in Bangladesh [16,17] exposed to iAs above 10ug/L, the maximum contaminant level set by the U.S. Environmental Protection Agency (EPA) [18] and World Health Organization (WHO) [1,19]. Other sources of arsenic exposure include the consumption of contaminated seafood and rice [20–22].

iAs metabolism in humans is composed of a series of reduction and methylation reactions occurring primarily in the liver with some metabolism potentially occurring in other tissues such as the kidney [23,24]. iAs in the form of arsenite (iAs$^{III}$) or arsenate (As$^V$) enters the body. Based on the Challenger model of metabolism [14,23,25], iAs$^V$ can be reduced to iAs$^{III}$, which can be methylated in a reaction catalyzed by arsenic (+3 oxidation state)

T73M007281, The National Institute of
Environmental Health Sciences award number
5F30ES031858-02 (M.B.C.), Susan G. Komen
Research Training Grant under award number
GTDR16376189, and the National Institute of
Aging under award number T32AG51146-5. The
Genotype-Tissue Expression (GTEx) Project was
supported by the Common Fund of the Office of
the Director of the National Institutes of Health, and
by NCI, NHGRI, NHLBI, NIDA, NIMH, and NINDS.
The funders had no role in study design, data
collection and analysis, decision to publish, or
preparation of the manuscript.

**Competing interests:** The authors have declared
that no competing interests exist.

methyltransferase (AS3MT) producing monomethyarsonic acid (MMA$^V$) [14], which can be reduced to monomethylarsonous acid (MMA$^{III}$). A second methylation step, also catalyzed by AS3MT, produces dimethylarsinic acid (DMA$^V$), which can be reduced to dimethylarsinous acid (DMA$^{III}$), the end-product of iAs metabolism in humans.

Individuals' arsenic metabolism efficiency (AME) is often represented as the percentage of each arsenic species in urine relative to all species (iAs%, MMA%, and DMA%) [7,14], with higher DMA% indicating more efficient metabolism. While age, sex, and environmental factors contribute to inter-individual variation in AME [14,26], inherited genetic variation also plays an important role. Variation in the *AS3MT*/10q24.32 region has shown clear association with AME in multiple populations, including Bangladesh and American Indian communities in the US, with multiple independent association signals identified [14,27–29]. Prior studies have found that higher levels of urinary MMA% and/or lower levels of urinary DMA% are associated with increased risk for cancer, cardiovascular disease [30] and arsenic-induced skin lesions [14]. AME-associated SNPs have also been shown to impact the risk of arsenic-induced skin lesions, reflecting increased arsenic toxicity among those with lower AME [31].

The causal variants underlying the observed associations between *AS3MT*/10q24.32 variation and AME remain unknown. Previous studies have had limited SNP density, small sample sizes, and have focused on single populations. In this work, we generate targeted sequencing data for multiple arsenic-exposed populations to identify candidate causal variants underlying the association between 10q24.32 variants and AME. We perform in-silico functional annotation to assess potential functional impact and further prioritize potential causal variants. Finally, we conduct co-localization analyses to examine variants' potential effects on *AS3MT* expression and/or nearby DNA methylation. This work enhances our understanding of the genetic mechanism underlying variation in AME and susceptibility to iAs toxicity.

## Results

### Arsenic exposure and AME

Total urinary arsenic varied substantially across cohorts, with higher concentrations in HEALS compared to SHS and NHSCS (**Fig 1**). This difference was also reflected in measures of arsenic in participants' drinking water (**S1 Fig**). DMA% was highest in NHSCS and lowest in HEALS (**Table 1** and **S2 Fig**). MMA% was highest in SHS and lowest in NHSCS while iAs% was highest in HEALS and lowest in NHSCS (**Table 1**).

### Association analyses

Conditional association analysis of TOPMed-imputed data identified association signals for urinary DMA% in the 10q24.32 region for all three cohorts (**Fig 2**). We identified three independent signals in HEALS (lead SNPs rs145537350, rs12573221, and rs4919687), two in SHS (rs191177668 and rs4919688), and one in NHSCS (rs191177668). The per-allele association estimates for these lead SNPs varied in magnitude from ~2% to ~12% (in DMA% units) (**Table 2**). These results are similar to those observed for the non-imputed data (**S3 Fig**). Analysis of MMA% and iAs% across all three cohorts produced results generally consistent with those observed for DMA%, with similar association signals detected for all three arsenic species (**S4, S5, S6** and **S7 Figs**). Alleles associated with increased DMA% tended to be associated with decreased MMA% and iAs%.

SHS and NHSCS share a lead SNP (rs191177668) and this SNP is in LD ($r^2$ = 0.5–1.0 depending on reference population) with a HEALS lead SNP, rs145537350 (**Fig 3**), suggesting a shared causal variant across cohorts. The shared lead SNP in SHS and NHSCS, rs191177668, has a p-value of $8 \times 10^{-8}$ and beta estimate of -0.062 in HEALS (isolated primary signal).

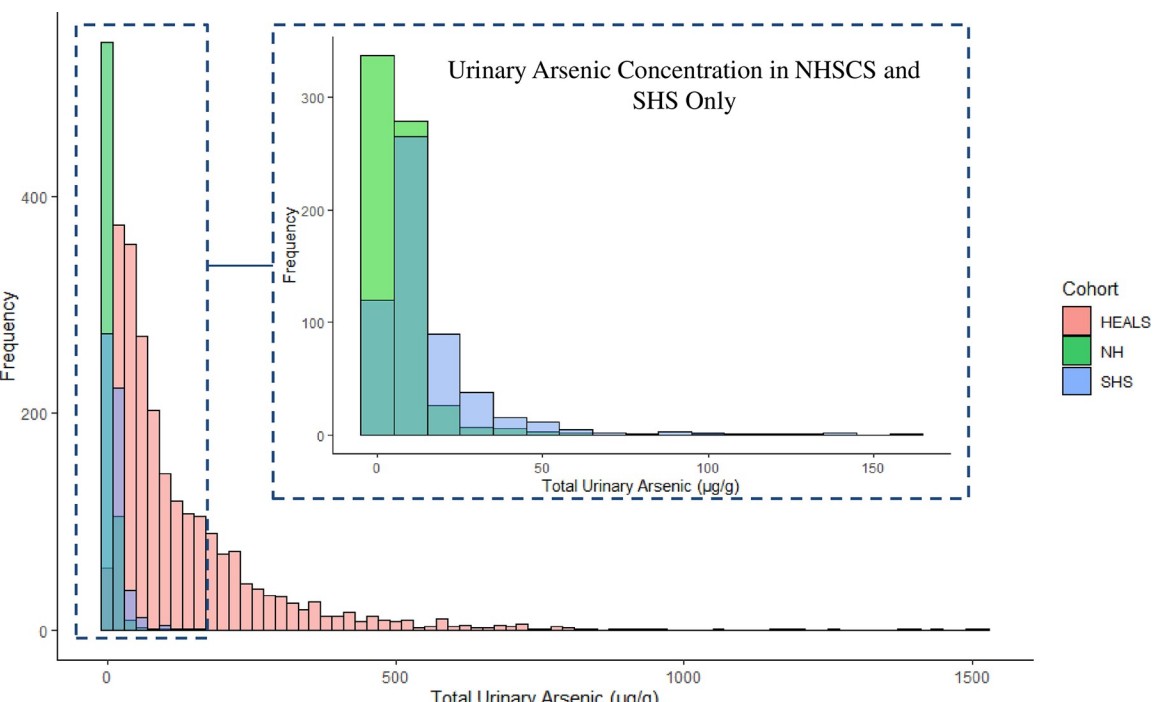

**Fig 1. Urinary arsenic concentrations in HEALS, NHSCS, and SHS.** Distributions of total urinary arsenic (µg/g of creatinine) are shown for three arsenic-exposed populations: The Health Effect of Arsenic Longitudinal Study (HEALS, in red), the New Hampshire Skin Cancer Study (NHSCS, in green), and the Strong Heart Study (SHS, in blue). The inset Fig shows SHS and NHSCS only.

Controlling for the primary signal, this SNP has p-values of 0.6 and 0.2 in the HEALS secondary and tertiary analyses. We also observe two signals in HEALS and one in SHS that appear distinct (or population-specific), with minimal LD among the lead SNPs (**Fig 3**). Some lead SNPs differ substantially in MAF across populations (**Table 3**). For example, SHS lead SNP rs145537350 is common in SHS (MAF = 0.14) but has much lower frequency in HEALS (0.007) and NHSCS (0.005). We further examined MAF differences for the lead SNPs across SHS centers (**S1 Table**). Here, we find generally consistent MAF across SHS centers, with the exception of rs4919688 which is more common in the Arizona center (MAF = 0.482) compared with the Dakotas and Oklahoma centers (0.252, 0.221 respectively).

We examined the impact of BMI as a covariate in each cohort. While BMI is associated with DMA% in each cohort, the association for the lead SNPs identified in each cohort are largely unchanged. In the cohorts with available measurements of arsenic in drinking water (HEALS and NHSCS) water arsenic exposure was inversely associated with DMA% (**S8 Fig**), as reported previously [14,32]. However, there was no detectable interaction (P<0.05) between any of the lead SNPs and exposure level measured in drinking water.

## Fine-mapping

The results of ancestry-specific fine-mapping of individual-level data performed using SuSiE [33,34]] were consistent with those observed in our association analyses, with three distinct 95% confidence sets identified in HEALS and two in SHS (**Fig 4**). We find overlapping SNPs in the confidence sets for HEALS set 1 and SHS set 1 (**S2, S3, and S4** Tables). For SHS, the confidence sets identified through the analysis of summary statistics were somewhat different than those obtained from analyses of individual-level data, with SNPs in summary statistic based SHS confidence set 1 overlapping with SNPs in HEALS sets 1 and 3 (**S2–S5** Tables).

**Table 1. Participant characteristics and arsenic species concentrations stratified by cohort.**

| Characteristics | HEALS (n = 2428) | SHS (n = 557) | NH (n = 662) |
|---|---|---|---|
| Sex [n (%), male] | 1428 (58.8) | 229 (41.1) | 393 (59.4) |
| Age [mean ± SD] | 41.8 ± 10.5 | 56.2 ± 8.3 | 64.4 ± 7.6 |
| BMI (kg/m$^2$kg/m2 [n (%)]) | | | |
| <18.5 | 1043 (43.0) | 5 (0.9) | 7 (1.1) |
| 18.5–24.9 | 1206 (49.8) | 87 (15.6) | 196 (29.6) |
| 25–29.9 | 137 (5.6) | 197 (35.4) | 250 (37.8) |
| >30 | 17 (0.7) | 267 (47.9) | 209 (31.6) |
| Smoking [n (%), has smoked] | 979 (40.3) | 218 (39.1) | 276 (41.7) |
| iAs% | | | |
| mean ± SD | 15.2 ± 6.5 | 8.8 ± 5.6 | 7.9 ± 5.2 |
| IQR | 11, 18.3 | 5.2, 10.8 | 4.4, 9.8 |
| Min, Max | 0, 70.3 | 0.62, 59.7 | 0.5, 35.7 |
| MMA% | | | |
| mean ± SD | 13.7 ± 5.2 | 14.6 ± 5.6 | 10.4 ± 4.5 |
| IQR | 9.8, 16.7 | 10.5, 17.6 | 7.3, 13.2 |
| Min, Max | 0, 34.7 | 0.53, 45.9 | 0.6, 36.7 |
| DMA% | | | |
| mean ± SD | 71.2 ± 8.7 | 76.7 ± 9.1 | 81.8 ± 8.1 |
| IQR | 66.1, 77.2 | 72.1, 83 | 77, 87.1 |
| Min, Max | 27.4, 92.9 | 32.4, 94.5 | 35.1, 97.7 |
| Arsenobetaine (ug/L) | | | |
| mean ± SD | 1.5 ± 3.5 | 2.9 ± 8.1 | 35.9 ± 111.9 |
| IQR | 0, 1.8 | 0.43, 1.7 | 1.4, 26.6 |
| Min, Max | 0, 85.19 | 0.07, 97 | 0.07, 1782.2 |
| Total Urinary Arsenic (ug/g) | | | |
| mean ± SD | 138.5 ± 164.7 | 15.2 ± 18.8 | 7.3 ± 8.96 |
| IQR | 38.2, 174.7 | 5.8, 16.8 | 3.3, 8.3 |
| Min, Max | 3.7, 1528.5 | 0.54, 161.95 | 0.71, 111.6 |

Abbreviations: iAs%, percentage of inorganic arsenic; MMA% percentage of monomethylarsonic acid; DMA%, percentage of dimethylarsinic acid; IQR, interquartile range

However, in both the summary statistic and individual-level data analyses, no overlap was observed between HEALS confidence set 2 and any SHS set or between SHS set 2 and any HEALS set. SuSiE did not identify a confidence set for NH, potentially due to the weaker associations observed in this group.

We examined the shared signal through meta-analysis of the isolated HEALS and SHS association signal results using MANTRA. The 95% confidence set produced contained 8 variants including the lead primary SNPs in HEALS (rs145537350) and SHS (rs191177668) which had posterior inclusion probabilities of 0.488 and 0.251, respectively (S3 Table). This confidence set also included the lead tertiary HEALS SNP (rs4919687), a variant included in HEALS set 3 and in the SHS summary statistic-based set 1.

## In-silico functional annotation

We aligned credible set variants with markers of open chromatin (Dnase I), cis-Credible Regulatory Elements (CREs), H3K27Ac, H3K4me3, and transcription factor (TF) binding sites (specifically TFs in cells and tissue types related to the kidney, liver, and heart). Three variants

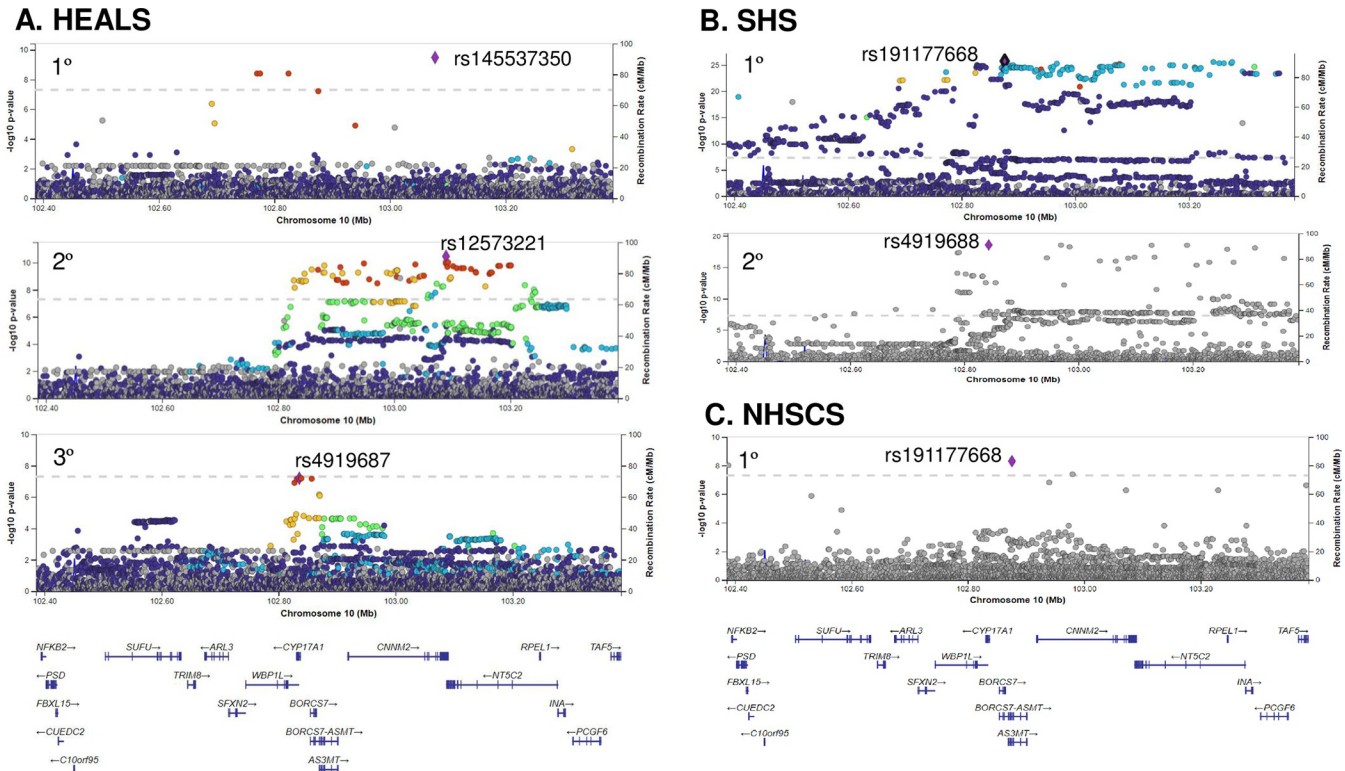

**Fig 2. Isolated association signals for DMA% in the 10q24.32 region in three arsenic-exposed populations.** P-values are from linear models adjusted for age, sex, and population structure (in HEALS and SHS). The top panel for each population shows the primary association (adjusted for lead SNPs from all non-primary signals, if present). Additional panels show p-values for secondary (HEALS and SHS) and tertiary (HEALS) association signals (adjusted for lead SNPs from all other signals). Three signals were identified for HEALS, two for SHS, and one for NHSCS. LD estimates are based on several 1,000 Genomes populations (SAS for HEALS, MXL/PUR/CLM/PEL for SHS, and EUR for NHSCS).

**Table 2. Lead SNPs for isolated DMA% association signals by cohort.**

| Population | Lead SNP | Confidence Set Membership | MAF | Imputed/Genotyped | Beta[a] | P-Value |
|---|---|---|---|---|---|---|
| HEALS | | | | | | |
| | rs145537350 | HEALS CS 1 | 0.007 | Genotyped | -0.1197 | $5.3 \times 10^{-16}$ |
| | rs12573221 | HEALS CS 2 | 0.134 | Genotyped | 0.0264 | $2.2 \times 10^{-13}$ |
| | rs4919687 | HEALS CS 3 | 0.122 | Genotyped | -0.0199 | $6.4 \times 10^{-8}$ |
| SHS | | | | | | |
| | rs191177668 | SHS CS 1 | 0.143 | Imputed | -0.0712 | $1.8 \times 10^{-26}$ |
| | rs4919688 | SHS CS 2* | 0.268 | Imputed | -0.0428 | $2.8 \times 10^{-19}$ |
| NH | | | | | | |
| | rs191177668 | NA | 0.012 | Imputed | -0.0947 | $5.0 \times 10^{-9}$ |

Abbreviations: SNP, single nucleotide polymorphism; MAF, minor allele frequency; HEALS, Health Effects of Arsenic Longitudinal Study; SHS, Strong Heart Study; NH, New Hamphire Skin Cancer Study; CS, Confidence Set. A Models were adjusted for age and sex.

Models in SHS additionally included top 5 genotyping PCs, and models in HEALS were adjusted for kinship between participants. Isolated, independent signals were identified by conditional association models adjusted for other signals identified in the region. Reported beta values reflect the change in DMA% scales from 0–1.

*Refers to the association-based SHS confidence set

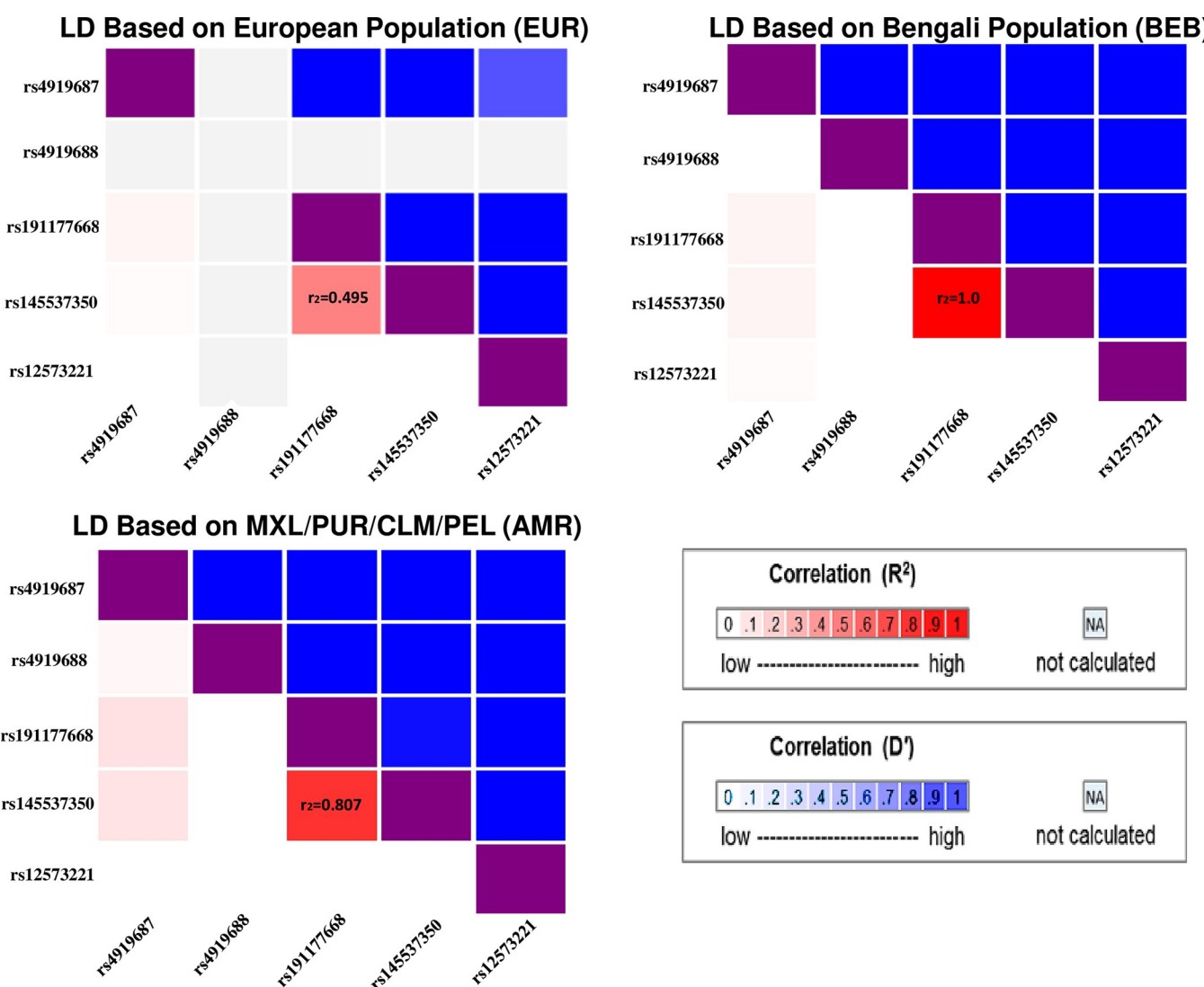

**Fig 3. Linkage Disequilibrium (LD) among DMA% lead SNPs in multiple reference populations.** All LD estimates are based on 1,000 Genomes data obtained from LDLink. We observe strong LD (r2 = 0.495–1.0) between rs145537350 (HEALS) and rs191177668 (NHSCS and SHS), but weak LD among all other variants.

**Table 3. Minor Allele Frequency (MAF) comparison of lead signals across arsenic-exposed cohorts.**

| Lead SNP | Population Identified In | Confidence Set Membership | HEALS MAF | SHS MAF | NH MAF |
|---|---|---|---|---|---|
| rs145537350* | HEALS | HEALS CS 1 & SHS CS 1 | 0.007 | 0.143 | 0.005 |
| rs12573221 | HEALS | HEALS CS 2 | 0.134 | 0.002 | 0.025 |
| rs4919687 | HEALS | HEALS CS 3 | 0.122 | 0.220 | 0.267 |
| rs191177668* | SHS & NH | SHS CS 1 | 0.012 | 0.143 | 0.012 |
| rs4919688 | SHS | SHS CS 2** | 0.026 | 0.268 | 0.003 |

Abbreviations: SNP, single nucleotide polymorphism; MAF, minor allele frequency; HEALS, Health Effects of Arsenic Longitudinal Study; SHS, Strong Heart Study; NH, Hew Hampshire Case-Control Study of Squamous Cell Carcinoma

*Indicates SNPs that are in high LD with each other across multiple reference population including Bengali, Ad-Mixed Americans, and Europeans.

** Refers to the association-based SHS confidence set

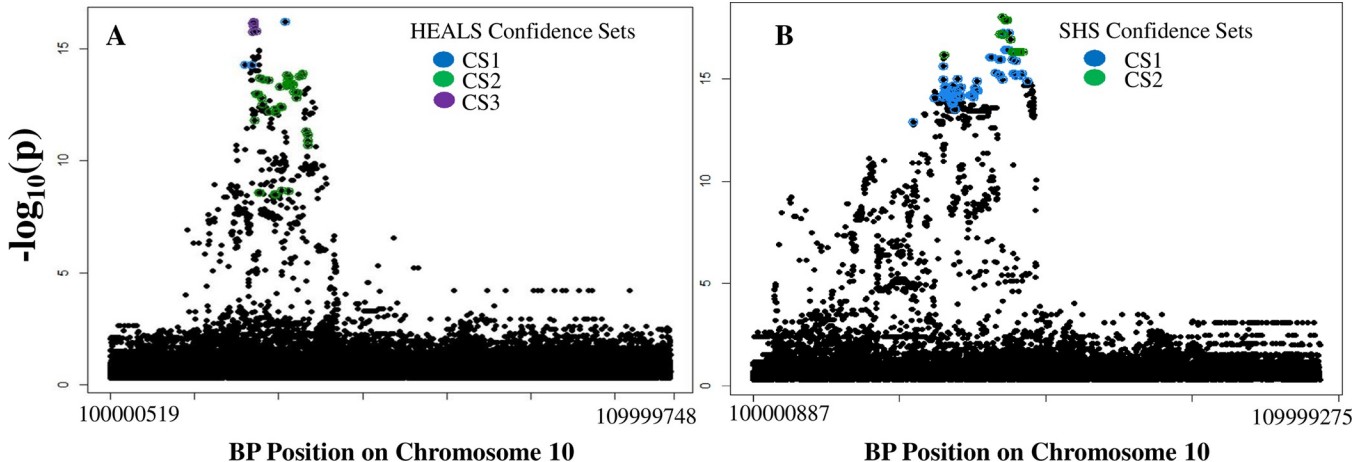

**Fig 4. SuSiE Fine-Mapping results.** 95% confidence sets for HEALS (A) and SHS (B) are each highlighted in different colors and mapped onto the overall DMA% association results.

in HEALS confidence set 1 (**Table 4** and **S9 Fig**), one variant in HEALS set 3 (**Table 4** and **Fig 5**), and 12 of the 50 variants in the HEALS set 2 overlapped at least one of the examined features. Examining the 85 variant SHS confidence set that contains the shared lead variant in SHS and NHSCS (rs191177668) as well as the lead variant in HEALS (rs145537350), we find that 24 of the variants overlap functional features (**S6 Table**).

## Cis-eQTL identification and co-localization analysis

*AS3MT* is expressed in most human tissue types, and this expression is highly variable ranging from 411.5 TPM in the adrenal gland to 0.92 in whole blood (**S10 Fig**). We identified cis-eQTLs for *AS3MT* in 45 tissue types (among 47 analyzed), with 27 tissue types having multiple

**Table 4. In-Silico functional examination of HEALS confidence set variants.**

| Variant | Coordinates | PIP | Candidate cis-Regulatory Element (cCRE) | Transcription Factors binding the cCRE | Histone Marks at the cCRE | Classification |
|---|---|---|---|---|---|---|
| **Confidence Set 1** | | | | | | |
| rs142093276 | 10:102769853 | 0.051 | EH38E1495376 | 32 | 13 | Distal enhancer-like signature |
| rs17114969 | 10:102775204 | 0.051 | EH38E1495388 | 14 | 27 | Proximal enhancer-like signature |
| rs4919681* | 10:102824339 | 0.051 | None | | | |
| **Confidence Set 3** | | | | | | |
| rs4919684* | chr10:102827267 | 0.083 | None (3 within 2kb) | | | |
| rs10883783* | chr10:102831395 | 0.154 | None (3 within 2kb) | | | |
| rs743575* | chr10:102835149 | 0.189 | EH38E1495455 | 34 | 14 | Distal enhancer-like signature |
| rs4919687* | chr10:102835491 | 0.143 | None (4 within 2kb) | | | |
| rs10883784* | chr10:102838165 | 0.160 | None (4 within 2kb) | | | |
| rs10786714* | chr10:102838849 | 0.160 | None (4 within 2kb) | | | |
| rs4919690 | chr10:102856743 | 0.087 | None (3 within 2kb) | | | |

Abbreviation; PIP, posterior inclusion probability.

* Indicates SNPs also present in SHS confidence sets

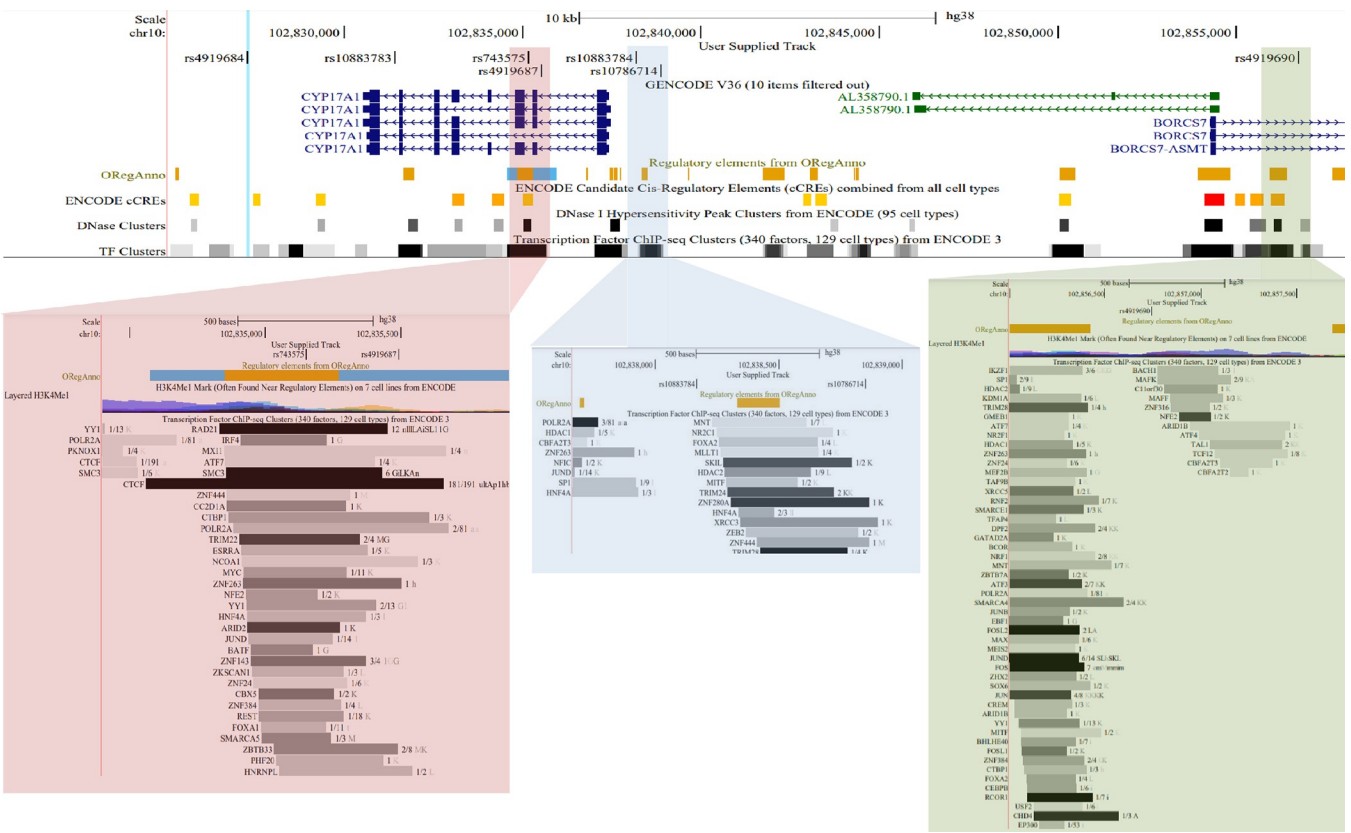

**Fig 5. In silico functional annotation of candidate causal variants of HEALS confidence set 3.** Candidate causal SNPs from HEALS confidence set 3 (lead SNP rs4919687) overlap with genomic features including candidate cis-regulatory elements and transcription factor binding sites. Highlighted panels show details of candidate causal SNPs overlapping these features. Note that *AS3MT* is located downstream of *BORCS7* and overlaps *BORCS7-AS3MT*.

*AS3MT* cis-eQTLs (as many as four). Following identification of these QTLs, we wanted to determine if any eQTLs shared a common causal variant with our AME association signals thereby representing a regulatory mechanism by which our identified SNPs impact AME. Cis-eQTLs in 22 tissue types had a lead eSNP in high LD ($r^2$>0.7) with rs4919687, a lead DMA% SNP in HEALS found in confidence set 3. Among these, rs4919690 (in confidence set 3) was the lead eSNP in 14 tissue types. Eight tissue types had a lead eSNP in high LD with rs12573221 (lead SNP in HEALS confidence set 2) and 5 tissues had a lead eSNP in high LD with rs145537350 (lead SNP in HEALS set 1) and rs191177668 (shared lead SNP in SHS and NHSCS). Examining the individual lead SNPs associated with urinary DMA% in the context of our eQTL analysis, we found that HEALS lead SNP rs4919687 had p < $5x10^{-8}$ in forty tissue types and the shared lead SNP in SHS and NHSCS, rs191177668, had p-value < $5x10^{-8}$ in eight tissue types (based on primary eQTL analysis). The second lead SNP in SHS (rs4919688) was not identified in any of our QTL analyses.

Following the identification of eQTLs in high LD with our AME-associated SNPs, we performed a co-localization analysis to determine whether the same causal variants impact AME and *AS3MT* expression, providing a regulatory mechanism by which our identified SNPs influence AME. Under the assumption that 50% of DMA% SNPs are eQTLs ($p_{12} = 5x10^{-6}$), we found evidence for co-localization between *AS3MT* cis-eQTLs and the DMA% signal represented by rs4919687 in 21 tissue types (PP of common causal variant (CCV) >80%) (**Table 5**). The rs4919687 allele associated with decreased AME (lower DMA%) was associated with

**Table 5. Co-localization of AS3MT eQTLs identified in GTEx tissues with the association signal for DMA% (lead SNP rs4919687) identified in a Bangladeshi population across a range of prior probabilities.**

| Tissue | eQTL Lead SNP | PP of Co-localization Under Different Assumptions | | | | Adjustments to Isolate eQTL signal |
|---|---|---|---|---|---|---|
| | | 50% of DMA% SNPs are eQTLs | 25% of DMA% SNPs are eQTLs | 10% of DMA% SNPs are eQTLs | %5 of DMA% SNPs are eQTLs | |
| Brain: Hippocampus | rs4919690 | 0.97 | 0.93 | 0.82 | 0.68 | 0 SNPs |
| Brain: Nucleus Accumbens Basal Ganglia | rs10883784 | 0.97 | 0.92 | 0.80 | 0.66 | 0 SNPs |
| Testis | rs4919690 | 0.91 | 0.78 | 0.55 | 0.37 | 0 SNPs |
| Vagina | rs12775431 | 0.80 | 0.58 | 0.32 | 0.18 | 0 SNPs |
| Artery Aorta | rs4919690 | 0.99 | 0.98 | 0.94 | 0.88 | 2˚ SNP |
| Artery Coronary | rs4919690 | 0.97 | 0.92 | 0.79 | 0.64 | 2˚ SNP |
| Brain Cortex | rs4919690 | 0.98 | 0.96 | 0.88 | 0.78 | 2˚ SNP |
| Mammary Tissue | rs12416687 | 0.94 | 0.84 | 0.64 | 0.46 | 2˚ SNP |
| Colon Transverse | rs11191421 | 0.45 | 0.23 | 0.09 | 0.04 | 2˚ SNP |
| Heart: Atrial Appendage | rs12416687 | 0.98 | 0.94 | 0.84 | 0.72 | 2˚ SNP |
| Heart: Left Ventricle | rs11191436 | 0.93 | 0.82 | 0.61 | 0.42 | 2˚ SNP |
| Liver | rs4919690 | 0.91 | 0.78 | 0.54 | 0.36 | 2˚ SNP |
| Esophagus Muscularis | rs4919690 | 0.97 | 0.93 | 0.82 | 0.68 | 1˚ SNP |
| Adipose Visceral Omentum | rs4919690 | 0.99 | 0.97 | 0.92 | 0.84 | 2˚, 3˚ SNP |
| Anterior Cingulate Cortex | rs4919690 | 0.98 | 0.96 | 0.88 | 0.77 | 2˚, 3˚ SNP |
| Colon Sigmoid | rs4919690 | 0.97 | 0.91 | 0.77 | 0.62 | 2˚, 3˚ SNP |
| Lung | rs4919690 | 0.99 | 0.98 | 0.93 | 0.86 | 2˚, 3˚ SNP |
| Nerve Tibial | rs4919690 | 0.99 | 0.98 | 0.94 | 0.88 | 2˚, 3˚ SNP |
| Muscle Skeletal | rs12416687 | 0.87 | 0.71 | 0.45 | 0.28 | 1˚, 2˚ SNP |
| Skin Suprapubic | rs1475642 | 0.96 | 0.90 | 0.74 | 0.58 | 1˚, 2˚ SNP |
| Gastroesophageal Junction | rs4919690 | 0.97 | 0.91 | 0.76 | 0.61 | 1˚, 3˚ SNP |
| Adipose Subcutaneous | rs4919690 | 0.98 | 0.93 | 0.83 | 0.69 | 2˚,3˚,4˚ SNP |
| Artery Tibial | rs12416687 | 0.98 | 0.93 | 0.82 | 0.69 | 2˚,3˚,4˚ SNP |

lower *AS3MT* expression across all tissue types in which co-localization was observed (**Fig 6B**), consistent with a mechanism in which lower *AS3MT* mRNA levels result in lower protein levels and lower enzymatic activity; thereby decreasing AME.

Varying the prior probability ($p_{12}$) of the percentage of DMA% SNPs that are also cis-eQTLs from 50% to 5% decreased the number of tissue types in which co-localization was observed (PP of CCV>80%) between *AS3MT* cis-eQTLs and rs4919687 from 21 to 4 (**Table 5 and Fig 6A**) and resulted in no co-localization between *AS3MT* cis-eQTLs and other DMA% signals (other than HEALS set 3).

Considering the additional lead signals identified in our population-specific analyses, we observed co-localization between DMA% signal rs12573221 (HEALS set 2) and an *AS3MT* eQTL in aortic artery. Even under the most liberal priors, we observed no evidence of co-localization for DMA% signals represented by rs145537350 (HEALS set 1) or rs191177668 (shared SNP in SHS and NHSCS).

Beyond *AS3MT*, we examined 27 additional genes within 500kb of *AS3MT*. HEALS DMA% lead SNPs rs4919687 (HEALS set 3) was in high LD with eQTLs for multiple genes (in at least one tissue type), including *CYP17A1OS*, *AL356608.1*, *CYP17A1*, *BORCS7*, *NT5C2*, and *WBP1L*. Among these, a *BORCS7* eQTL (represented by rs11191421 and rs4919690) present in multiple tissues showed strong LD with HEALS lead SNP rs4919687 (set 3) in 43 tissue types. We also observed two genes, *NFKB2* and *RPARP-AS1*, with cis-eQTLs whose lead SNPs

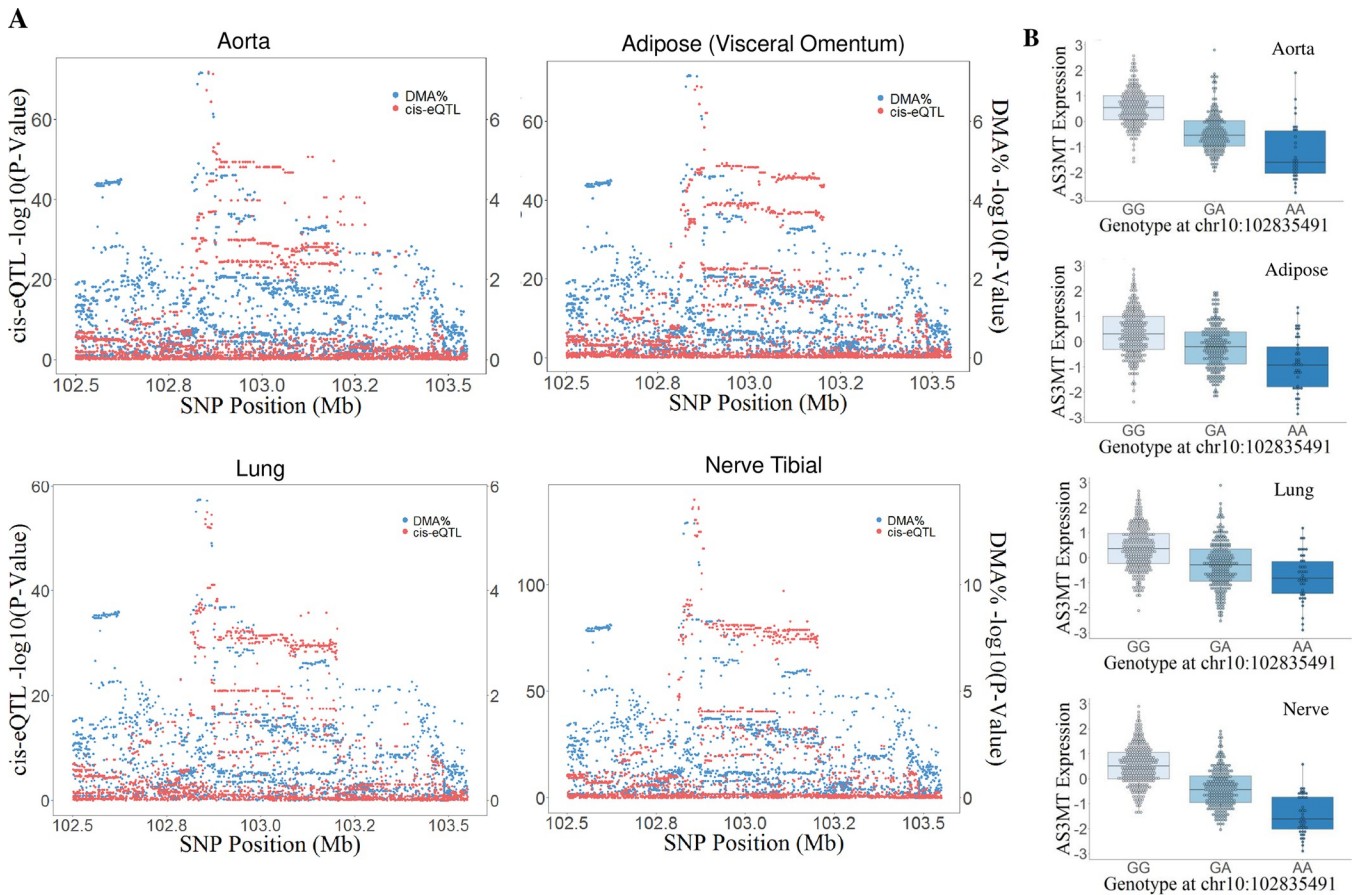

**Fig 6. Co-localization between HEALS DMA% association signal (rs4919687) and *AS3MT* eQTLs in multiple tissues.** We observe evidence of co-localization in 21 GTEx tissue types, with four example tissue types shown in Panel A (aorta, adipose-visceral omentum, lung, and nerve-tibial). The low efficiency (low DMA%) allele at rs4919687 is consistently associated with lower *AS3MT* expression in all tissues for which co-localization was observed.

were in high LD with a HEALS DMA% lead SNP rs12573221 (set 2) in tibial nerve and transverse colon tissues respectively. Finally, we observed one gene, *SUFU*, with a cis-eQTL in high LD with a DMA% association signal from SHS (rs191177668, association-based set 1) in the tibial artery.

Under the assumption that 50% of DMA% SNPs are cis-eQTLs ($p_{12} = 5 \times 10^{-6}$), we identified 25 tissue types in which a cis-eQTL for *BORCS7* co-localized with the DMA% signal represented by rs4919687. Thus, both *AS3MT* and *BORCS7* eQTLs co-localize with this DMA% signal. We also observed co-localization between this DMA% signal and tissue-specific eQTLs in *CYP17A1OS* (thyroid) and *CYP17A1* (frontal cortex of the brain) (PP>80%) (**S7 Table**). Similar to *AS3MT* results, reducing $p_{12}$ (to a 5% probability of co-localization), resulted in much lower probabilities of co-localization for *BORCS7* (**S7 Table**).

## Cis-mQTL identification and co-localization analysis

Among the nine tissue types analyzed, we identified mQTLs in strong LD with rs4919687 (HEALS set 3) in five tissue types: transverse colon, kidney cortex, lung, ovary, and testis. We also identified mQTLs in high LD with rs12573221 (HEALS set 2) in five tissue types: breast (mammary), lung, skeletal muscle, prostate, and testis. Finally, we identified mQTLs in high

LD with rs145537350 (HEALS set 1) in four tissue types: breast (mammary), kidney cortex, lung, and whole blood.

Under the assumption that 50% of DMA% signals are mQTLs ($p_{12} = 5 \times 10^{-6}$), we observed co-localization between DMA% signal rs4919687 and mQTLs in two tissue types: colon-transverse (one CpG) and ovary (two CpGs) (**S8 Table**). For colon, the associated CpG was in the gene body of *CYP17A1* while for ovary, both associated CpGs were in the gene body of *AS3MT*. We did not observe any co-localizations between mQTLs and the DMA% association signals represented by either rs12573221 (HEALS set 2) or rs145537350 (HEALS set 1).

We observed co-localizations between *AS3MT* cis-eQTL (rs4919690) and cg08650961 cis-mQTL (rs4919690) under the assumption that 50% of mQTLs are also eQTLs ($p_{12} = 5 \times 10^{-4}$) and the assumption that 25% of mQTLs are also eQTLs ($p_{12} = 2.5 \times 10^{-4}$).

### Effect modification analysis

Using known associations between individual characteristics and DMA%, we performed a series of effect modifications analyses to understand the interactions between these variables and our identified variants. We found no interaction between sex and the genetic effects on DMA% (S9 Table); the effects of our identified variants were similar in males and females. We also examined the interaction between smoking and DMA% and observed evidence of interaction between current smoking and the lead SNP in NHSCS ($p = 0.04$), but not in HEALS or SHS.

### Sensitivity analysis

Arsenobetaine, a form of organic arsenic found in seafood (see methods) was correlated with DMA% in all three cohorts (correlation of 0.18, 0.12, and 022 in HEALS, SHS, and NHSCS respectively); however, no clear differences were seen between our initial analysis results and those from a model that included adjustment for arsenobetaine. To further confirm our results, we performed a sensitivity analysis, excluding individuals with DMA% >85 (76 in HEALS, 224 in NHSCS, and 169 in SHS) to avoid the inclusion of DMA measures strongly influenced by exposure to organic arsenic. Here, we found no significant change in results with our lead SNPs still appearing among the top SNPs in the association analyses of all three populations.

### Discussion

The relationship between *AS3MT* genotype, gene expression in the 10q24.32 region, and arsenic metabolism are well-established; however, questions remain regarding the precise causal variants driving these associations, the potential differences in associations across ancestry groups, as well as the specific genes in the region influenced by 10q24.32 genotypes across different tissue types. In this project, we applied fine-mapping approaches to sequencing-based genotype data in the 10q24.32 AME-associated region to identify candidate causal variants across three cohorts exposed to varying levels of arsenic in their drinking water. Fine-mapping analyses revealed that there are likely multiple causal variants in the 10q24.32 region impacting AME (represented by DMA%), with at least one causal variant likely shared across populations. *In silico* functional annotation and QTL co-localization further revealed that several of our candidate causal variants overlap regulatory features and impact expression of *AS3MT* and local DNA methylation.

Under the assumption of shared biological mechanisms and shared causal variants across populations, cross-population association analyses can narrow the list of potential causal variants in a region by identifying SNPs showing consistent evidence of association across all

examined populations [35–37]. However, this was not the case in our study, as meta-analysis of shared association signals in HEALS and SHS produced a confidence set of 8 variants, a set larger than the corresponding confidence sets based on HEALS alone (4 and 7 variants). This is likely due to the broader signal observed in SHS, resulting in a confidence set with many more SNPs than those observed for HEALS (the result of more extensive LD among nearby variants in SHS). This extensive LD in SHS makes discriminating between potential causal variants more challenging in the meta-analysis context.

A novel finding of our analysis is the identification of associations that appear to be population-specific in both HEALS and SHS, which require further examination. Failure to replicate genetic associations across populations has been observed in many studies and has driven the increased emphasis on diversity in GWAS [38–43]. One explanation for a lack of replication is differences in allele frequencies, which can reduce power in populations with low MAF [34]. For example, one of the HEALS confidence sets (lead SNP rs12573221, confidence set 2) has a MAF range of 6.1–16.5% in HEALS and 0.1–4.4% in SHS; thus, the SNPs are likely too rare in SHS to be examined at the present sample sizes. Similarly, signal 2 (lead SNP rs4919688, confidence set 2) has a MAF of 26.8% in SHS and 2.6% in HEALS, which may explain the lack of replication of this SHS signal in HEALS. Our study used different sample sizes for our three populations. Larger sample sizes, as seen in HEALS, increase the study power and decrease the standard error of effect estimates, and thus improve our ability to identify association signals [44]. It is possible that we simply lacked the power to identify all association signals across all populations.

Beyond MAF, differences in LD likely contribute to the differences in observed signals across populations. HEALS signal three is represented by rs4919687 which has a MAF >0.1 in all populations. However, while this variant represents an independent candiate causal variant and confidence set (set 3) in HEALS, it co-occurs with SNPs from HEALS confidence set 1 in SHS confidence set 1 (derived from summary statics). Furthermore, we find the HEALS tertiary lead SNP (in HEALS confidence set 3) among the top 30 SNPs of the SHS primary association signal. Thus, a single confidence set in SHS (set 1) may capture two confidence sets in HEALS. Differences in LD patterns across the populations is a possible explanation for these observations [35,38,45], causing two distinct signals (in HEALS) to be indistinguishable in SHS.

In addition to these factors, differences in subject recruitment and inclusion among the studies used for this work may also have contributed to the differences in association signals. NHSCS included individuals both with and without skin cancer. Arsenic exposure and AME are risk factors for skin cancer, so this selection could contribute to biases in observed associations [30,46]. Chronic arsenic exposure is also associated with increased health risks, so the long-standing exposure in HEALS may also distinguish it from the other populations and make direct comparison challenging [46]. It is possible that the observed population-specific signals reflect true differences in genetic effects across populations. Gene-gene or gene-environment interactions could result in different effect size estimates across populations [38,40]. Differences in exposure level across populations may also impact the observed associations [40,41,43]. For instance, attenuation of SNP effects in populations with low exposure may result in low power to detect association [40]. We examined the gene-environment interaction between our lead SNPs and arsenic exposure in HEALS and while arsenic exposure did have an independent effect on DMA%, there was no evidence of effect modification between any of our identified variants and arsenic exposure (**S10 Table**).

We sought to understand the regulatory mechanisms by which the causal variants impact gene function using co-localization analyses focused on *AS3MT* and surrounding genes and DNA methylation features. *AS3MT* is expressed in most tissues at detectable levels (S10 Fig),

allowing us to examine co-localization across a wide range of tissues. Expression is highest in the adrenal gland, potentially due to co-regulation with nearby *CYP17A1* which plays a role in steroid hormone formation or for protection against arsenic which can disrupt endocrine function [47,48]. While co-localization was not detected for adrenal tissue, it was detected in many tissue types with low expression (e.g. subcutaneous adipose). This suggests that our results were not driven by the variability in expression across tissues and that low expression levels did not prevent QTL detection.

The liver is the major site of arsenic metabolism, but we observed only suggestive evidence of co-localization in this tissue type (Table 5 and S11 Fig) We detected *AS3MT* eQTLs in liver (rs4919690, $p = 7.15 \times 10^{-8}$), but our relatively small sample size may have limited our statistical power to robustly identify co-localization across all sets of priors analyzed.

Despite the ancestry mismatch between our Bangladeshi participants and the GTEx donors, we found compelling evidence of co-localization between our DMA% association signal (represented by rs4919687, HEALS set 3) and a multi-tissue cis-eQTL for *AS3MT* in 21 tissue types. The minor allele at SNP rs4919687 was associated with decreased DMA% and decreased *AS3MT* expression in tissues in which co-localization was observed, supporting the hypothesis that decreased expression results in lower amounts of the enzyme and ultimately lower AME. Our co-localization results suggest that HEALS SNP rs4919690 may be the causal variant underlying this signal (Posterior Inclusion Probability, PIP = 0.09), as it is the lead eSNP for the co-localizing *AS3MT* eQTL in 14 tissue types, and it is the lead eSNP for co-localizing *BORCS7* eQTL in 9 tissue types. The repeated appearance of this SNP as a lead SNP in GTEx QTL analyses (genotyping based on whole-genome sequencing) suggests causality. Some co-localization was observed between *AS3MT* eQTLs and a second association signal in HEALS, suggesting that this variant may also impact arsenic metabolism by regulating *AS3MT* expression across tissues.

No prior evidence suggests a role for *BORCS7* in arsenic metabolism; however, we observe co-localization between a DMA% association signal and cis-eQTLs for *BORCS7* in multiple tissues (as observed previously [49]). Expression levels of *AS3MT* and *BORCS7* are correlated in nearly all tissues in which co-localization is observed, suggesting co-regulation by a common causal variant (and potentially other mechanisms) or two causal variants in very strong LD. The correlated expression of the two genes has been noted previously [50]. A mouse strain carrying a human *BORCS7/AS3MT* was created to study arsenic metabolism as the *AS3MT* promoter abuts the 3' UTR of *BORCS7* [51]. However, this study did not describe any specific role for *BORCS7* in arsenic metabolism. It is likely that the SNPs in this region are pleiotropic, influencing both *AS3MT* and the expression of the surrounding genes. Furthermore, co-regulation of these genes has been previously reported in multiple tissue types [31,49,50,52].

The co-localization of mQTLs with DMA% association signals further increases support for rs4919690 as a causal variant, as mQTLs for 10q24.32 CpGs co-localize with eQTLs (represented by lead SNP rs4919690) for *AS3MT* in lung tissue. The DMA% decreasing allele at rs4919690 (A) is associated with increased methylation of cg08650961, a CpG located in the body of the CNNM2 gene outside of a CpG island, and decreased *AS3MT* expression in lung tissue.

The population mismatch between our studies of AME (Bangladeshi and American Indian populations) and GTEx donors (primarily European ancestry) likely decreased our power to detect co-localization in both our eQTL and mQTL analysis. Furthermore, the smaller sample size of SHS likely increased the standard error of association estimates in this population, and when this is combined with the population mismatch between SHS and GTEx we likely had a decreased ability to detect co-localization in this population Our mQTL analysis sample sizes were also small, some n<100 samples, and our analyses were restricted to 9 tissue types with

available DNAm data. This limited our power for mQTL detection and our ability to detect co-localization. Additionally, in the populations examined for this study, we are not able to fully assess the contribution of organic sources of arsenic to variability in our AME phenotype, which could potentially bias the associations observed.

In this study, we do not assess the association of AME-related SNPs with arsenic toxicity risks. However, we have previously shown that the association signals represented by rs12573221 and rs4919687 show clear associations with arsenic-induced skin lesion risk in HEALS [53]. Similarly, the novel signal for SHS and NH that we report here (rs191177668), which is in strong LD with and can serve as a proxy for the novel signal in HEALS (rs145537350), is also associated with skin lesion risk in the dataset previously described (OR = 1.64, CI = 1.2, 2.24) [53].

This study builds on previous work in the 10q24.32 region [29,31,49,54,55] in several ways. Previous studies have found signs of positive selection near the *AS3MT* gene associated with efficient arsenic metabolism [56]. They found selection signals in multiple populations including several from South America and East Asia and found that several the SNPs associated with the protective haplotype were also associated with MMA%. None of the identified SNPs showed a significant association with DMA%, though in the context of our study one SNP did appear in lD with the secondary lead SNP in SHS.

We leverage data from multiple arsenic-exposed cohorts with diverse ancestry in a single study, allowing us to consider both shared and population-specific effects of inherited genetic variation on AME. Additionally, our targeted sequencing data enabled us to identify a novel, independent association between 10q24.32 variation and DMA% in HEALS that we were unable to detect in our previous array-based work [28]. We further increased our sample size for cis-eQTL analyses using the latest data from GTEx and incorporated both expression and methylation data into our examination of the SNPs' mechanism of action. Finally, we provide evidence that there are likely multiple causal SNPs within the 10q24.32 region associated with AME. Together, this allowed us to provide evidence regarding the potential causal variants and mechanisms underlying the established association between the 10q24.32 region and AME. Future studies can build on these findings, potentially establishing cellular or animal models with perturbations of potential causal sites in order to directly assess the impact of specific alleles on arsenic metabolism.

## Methods

### Ethics statement

The study protocol for the Health Effects of Arsenic Longitudinal Study (HEALS) was approved by the Institutional Review Boards of The University of Chicago, Columbia University, and the Bangladesh Medical Research Council. Details of the study were explained, and verbal informed consent was obtained from all participants. For the Strong Heart Study (SHS) the goals and procedures of the study were explained, and a signed consent form was obtained from each participant [57]. The study protocol was approved by the Indian Health Service Institutional Review Boards, and the participating communities [7,58]. The NHSCS was approved by the Committee for the Protection of Human Subjects of Dartmouth College. At enrollment, participants underwent a written, informed consent process [59].

### Study populations

This project leveraged data from three studies of arsenic-exposed individuals: The Health Effects of Arsenic Longitudinal Study (HEALS), the Strong Heart Study (SHS), and the New Hampshire Skin Cancer Study (NHSCS) (**Table 1**).

HEALS [17] is a prospective cohort study of a population from Araihazar, Bangladesh exposed to iAs through contaminated well water. 11,746 adults aged 18–75 years were recruited at baseline (1999–2001). All provided verbal informed consent. Water samples were collected from all 5,966 wells in the study area and tested for arsenic. In-person interviews, clinical evaluations, and blood and spot urine sample collection occurred in participants' homes by trained physicians using structured protocols. Follow-up in-person interviews were conducted biennially. The study protocol was approved by the Institutional Review Boards of The University of Chicago, Columbia University, and the Bangladesh Medical Research Council. Urinary arsenic species in baseline samples were previously measured for 4,794 participants by the Columbia University Trace Metals Core Laboratory. For this study, we included 2,426 individuals with both arsenic metabolite data and DNA available for targeted sequencing.

SHS [11,57,60] was established in 1989 to examine cardiovascular disease and risk factors among American Indian men and women. It includes participants from 12 tribes in the American Southwest, Northern Plains, and Southern Plains. Arsenic species were measured in 3,973 participants using procedures that have been described previously [61,62]. As some SHS participants are relatives, we restricted DNA for sequencing to 997 unrelated individuals with arsenic metabolite data and DNA available. We excluded 123 participants due to low coverage. An additional 6 were excluded due to missing arsenic species and/or covariate data. Therefore, our final dataset includes targeted sequencing for 868 individuals with arsenic species data.

NHSCS [63] is a population-based case-control study of squamous cell carcinoma of the skin. A total of 510 incident cases of histologically-confirmed, invasive squamous cell carcinoma were recruited and 483 NH residents, frequency matched on age and sex, were recruited as controls. Urine and home drinking water samples were collected and used to measure total water and urinary arsenic concentration urinary arsenic metabolites. Among the 993 participants, 288 lacked sufficient DNA and were excluded. This resulted in targeted sequencing for 706 participants with existing data on urinary arsenic metabolites.

## Measurement of urinary arsenic metabolites

Separation of arsenic species in urine samples from all three populations was performed using high-performance liquid chromatography [8,17,63]. This procedure was followed by IPC-MS to quantify arsenic species in urine samples. For this analysis, iAsIII and iAsV were summed to obtain total iAs, and each arsenic species (iAs, MMA, and DMA) is expressed as a percentage of the sum of each of these three species (iAs+MMA+DMA). Arsenocholine (AsC) and arsenobetaine (AsB) are nontoxic forms of (organic) arsenic and were excluded from our primary analyses. Details regarding the measurement protocols and the limit of detection for each metabolite used by each study have been described previously and can be found in S1 Text (Supplemental Methods) [17,54,58,63].

## The genotype-tissue expression project (GTEx)

SNP genotype and gene expression (RNA-seq) data from GTEx v8 [64,65] were used in this project, including data from 838 donors and 49 tissue types. The protocols for sequencing, processing, and quality control have been previously described [66,67].

## Targeted DNA sequencing

Protocols for the collection and extraction of DNA in HEALS, SHS, and NHSCS have been previously described [17,57,63]. Targeted sequencing for the 10q24.32 region was performed in all three cohorts using an Illumina TruSeq Custom Amplicon (TSCA) Kit designed in

Illumina's DesignStudio. Variant selection for targeted sequencing focused on variants in the 99% confidence set for each of the two AME association signals previously reported in HEALS [28] (>200 variants). In addition, variants in moderate LD ($r^2 > 0.4$ based on the 1000 Genomes (1KG) BEB population) with the lead variant of either association signal were also selected (42 and 208 variants for the primary and secondary signals respectively). An additional set of 527 variants was included to capture/tag all common variants in the 10q24.32 region. Coding regions of genes in the 10q24.32 region (*AS3MT*, *BORCS7*, *CNNM2*, *CYP17A1*, and *NT5C2*) were also selected to capture protein-coding variants. In total, 1,114 "targets" were selected (with targets corresponding to specific variants or exons of interest). The final TSCA design included 781 small regions across 1.5 Mb of the 10q24.32 region, each 400–450 bp in length. These amplicons covered 858 of our "targets" (with 256 targets being undesignable due to proximity to repetitive DNA sequences).

## Read alignment and genetic variant calling

Targeting sequencing data from all three cohorts (HEALS, SHS, and NH) were processed at the University of Chicago Bioinformatics Core using the Genome Analysis Toolkit (GATK) [68] Best Practices Workflow for germline short variant discovery [69]. For variant calling, raw paired-end reads were mapped to the hg19 reference using the Novoalign software. Variants were called for each sample using GATK HaplotypeCaller in GVCF mode to produce intermediate GVCFs which were then consolidated to a single GVCF file with the GenomicsDBImport tool. We used this consolidated file to perform joint variant calling across all samples by cohort, using GATK GenotypeGVCFs, which provided a set of raw SNPs and indels in VCF format. Biallelic SNPs with any of the following properties were excluded: QualbyDepth (QD) <2.0, FisherStrand (FS) >60.0, RMSMappingQuality (MQ) <40.0, StrandOddsRatio (SOR) >3.0, MappingQualityRankSumTest (MQRankSum) <-12.5, or ReadPosRankSum <-8.0. Additionally, indels with one or more of the following properties were excluded: QD <2.0, FS >200.0, ReadPosRankSum <-20, Inbreeding Coefficient <-0.8, or SOR >10.0. Finally, samples with low coverage, indicated by Depth of Coverage (DP) <30 were removed.

## Genotype quality control and imputation

Quality control (QC) of called SNPs was performed in PLINK v1.9 [70]. Beginning with 9,858 variants on chromosome 10 across all cohorts, QC included filtering by genotyping rate, sample missingness, and minor allele frequency (details in Supplemental Methods in S1 Text) and resulted in 455 10q24.32 variants in 2,357 samples in HEALS, 449 variants in 558 individuals in SHS, and 437 variants in 648 individuals in NHSCS. We imputed missing genotypes in the 10q24.32 region for each cohort using the TOPMed Imputation Server and the TOPMed reference panel [71], resulting in 36,468 10q24.32 variants in HEALS, 21,594 in SHS, and 28,401 in NHSCS.

## Association analysis

Data were analyzed using PLINK v1.9 [70] for SHS and NHSCS and using Genome-wide Complex Train Analysis (GCTA) [72] for HEALS to allow adjustment for cryptic relatedness using a genetic relationship matrix in a linear mixed model [28,31,54,72,73]. To generate this kinship matrix, we used genome-wide SNP data available for 2,434 HEALS participants. All regression models were adjusted for age and sex. Both sex and age are associated with AME; therefore, they were included as covariates in order remove variation in the outcome of interest and increase the power of our analysis. For SHS, we further included five SNP-based principal components (PCs) as covariates to account for population structure within the dataset

(PCs provided by SHS). PCs were not available for NHSCS due to lack of genome-wide SNP data. While the populations differ by weight, we do not perform association analyses across cohorts and therefore did not include BMI as a covariate in the models.

Our initial association analyses identified the SNP with the smallest P-value for each population (i.e., lead SNP). We then conditioned on the lead SNP(s) and ran a second association analysis for the region. We repeated this process of conditioning on identified lead SNPs until no additional, independent, signals were identified ($P<5x10^{-6}$). To enable downstream co-localization analyses, we isolated each identified association signal by including the lead SNPs for all other identified signals as covariates. Arsenobetaine is an organic form of arsenic that is highly stable and the major arsenic species found in most seafood [22]. Arsenobetaine is a marker of seafood intake, a diet that can also contain arsenosugars and arsenolipids, which were not measured but which have the potential to contribute to DMA% [22]. We therefore examined the relationship between arsenobetaine and DMA% and adjusted for arsenobetaine in multivariate models.

### Fine-mapping to identify candidate causal variants

We applied the Sum of Single Effects (SuSiE) [33] method (in R) to each cohort individually to identify confidence sets of candidate causal variants in each population. SuSiE uses an iterative Bayesian stepwise selection approach to produce credible sets of variables that are designed to be as small as possible while capturing the causal variant. SuSiE can take summary statistics as input or analyze individual-level data. Both options were used in this analysis to assess consistency. Under the assumption that causal variants are shared across populations, we can take advantage of population-specific patterns of linkage disequilibrium (LD) to narrow down the number of potential causal variants [45]. Therefore, we used MANTRA to meta-analyze the isolated association signals across cohorts [74,75].

### In-silico annotation of credible sets

Confidence set variants were examined using the UCSC Genome Browser [76] and ENCODE data [77]. We identified overlapping regulatory features (DNase hypersensitivity, histone marks, transcription factor binding sites, etc.) for all variants in each set.

### Identifying relevant eQTLs

We identified cis-eQTLs using expression data from 47 tissue types (cell lines excluded) from GTEx v8, with sample sizes ranging from 73 (Kidney Cortex) to 706 (Skeletal Muscle). We mapped cis-eQTLs for *AS3MT* and all genes within 500kb [64] using a series of linear regressions implemented in FastQTL [78] (previously described by GTEx and downloaded from https://github.com/francois-a/fastqtl). Regressions were adjusted for all covariates provided by GTEx for each tissue, including five genotyping PCs, sequencing platform and protocol, sex, and PEER factors (Probabilistic Estimation of Expression Residuals) which account for non-genetic factors that contribute to variation in gene expression [64,79].

To identify the eQTL(s) for a given gene, FastQTL finds the variant most strongly associated with gene expression (based on P-value). Once a lead SNP is identified, conditional analyses were used to identify additional independent cis-eQTLs and isolate each cis-eQTL signal (by adjusting for the lead SNPs for a given cis-eQTLs).

To identify eQTLs that may share a causal variant with a DMA% association signal, we identified cis-eQTL lead eSNPs in strong LD ($r^2 >0.7$) with a DMA% lead SNP. Due to the varied ancestries of our cohorts (and GTEx), we considered LD estimated from multiple 1KG groups (European (EUR), Bengali (BEB), and Admixed Americans (AMR)).

## Identifying relevant mQTLs

We used post-QC, inverse-normalized DNA methylation data generated for GTEx samples and post-QC genotype data from GTEx v8 [64,66,67,80]. Our analysis included a total of 856 samples from 367 donors [67]. The number of samples for each tissue type ranged from 42 to 190, and a total of 754,054 CpGs (measured using Illumina EPIC arrays) were examined across tissues [80]. The methods for mQTL identification have been described previously by Oliva et al [80]), and are the same as those used for our eQTL analysis and implemented in FastQTL [78] (https://github.com/broadinstitute/gtex-pipeline/tree/master/qtl). Multiple testing correction was performed [81] and conditional mQTL analysis was used to identify independent mQTLs using the same procedure as described for the eQTL analysis. We focused on mQTLs within the 10q24.32 AME-associated region (CpGs within 500 kb of *AS3MT*) and identified CpGs with mQTLs in high LD ($r^2 > 0.7$) with at least one of our DMA% association signals.

## Co-localization of DMA% and QTL signals

To determine if the observed DMA% association signals and the identified cis-eQTLs and mQTLs (from GTEx) share common causal variant(s), we conducted a Bayesian test for co-localization [81] using the coloc R package applied to all overlapping SNPs between the sets of summary statistics [82–84]. We applied coloc to our isolated DMA% association signals paired with the tissue type-specific, isolated cis-QTL association signals for all pairs for which the lead SNPs showed high LD ($r^2 > 0.7$) in a relevant 1KG population. The ancestry mismatch among GTEx (primarily European), HEALS (Bangladeshi), and SHS (American Indian) led us to examine LD estimates based on EUR, SAS, and AMR populations (for HEALS and SHS respectively) using data from the 1KG Project implemented in LDlink [85,86].

For co-localization, we specified a prior probability for a SNP being causal only for DMA% ($p_1$), only for gene expression or methylation ($p_2$), and for both DMA% and expression/methylation ($p_{12}$). We set the overall probability of being causal for DMA% ($p_1 + p_{12}$) as $10^{-5}$ and the overall probability of being an eQTL as $10^{-3}$ ($p_2 + p_{12}$), which is based on the average number of eGenes identified in GTEx and is typical in studies of complex traits and eQTLs. We then conducted a series of analyses in which the value of $p_{12}$ was varied to correspond to a 5%, 10%, 25%, and 50% probability that a causal variant for DMA% is also an eQTL. This corresponded to four values of $p_{12}$: $5 \times 10^{-7}$, $2 \times 10^{-6}$, $2.5 \times 10^{-6}$, and $5 \times 10^{-6}$. To assess co-localization between our tissue-specific, isolated mQTL signals and eQTL signals, we set the overall probability of being associated with an eQTL ($p_1 + p_{12}$) as $10^{-3}$ and the overall probability of being an mQTL as $10^{-3}$ ($p_2 + p_{12}$). We conducted a series of analyses in which the value of $p_{12}$ was varied to correspond to a 5%, 10%, 25%, and 50% probability that a causal variant for an eQTL is also an mQTL ($p_{12}$: $5 \times 10^{-5}$, $1 \times 10^{-4}$, $2.5 \times 10^{-4}$, and $5 \times 10^{-4}$).

Co-localization analyses produce a posterior probability (PP) of co-localization (H4), the PP that different causal variants underlie the two signals (H3), the PP there is an identifiable causal variant for DMA% but no detectable QTL signal (H1, and vice versa, H2), and the PP of no identifiable causal variant or QTLs (H0).

## Supporting information

**S1 Text.** a. Supplemental Methods: i. The Genotype-Tissue Expression Project. ii. Measurement of Urinary Arsenic Metabolites. iii. Quality Control for GTEx Whole Genome Sequencing Data. iv. Genotype Quality Control and Imputation. v. GTEx Expression Data Quantification. vi. Read Alignment and Genetic Variant Calling. vii. MANTRA meta-analysis. (PDF)

**S1 Fig. Arsenic Exposure Across Arsenic-Exposed Populations.** a. Measures of total water arsenic (µg/L) measured for individuals in three arsenic-exposed populations. The Health Effect of Arsenic Longitudinal Study (HEALS, in red), the New Hampshire Skin Cancer Study (NHSCS, in blue). Measurements in HEALS are based off arsenic levels in wells frequented by the individual participants; in NHSCS measurements were taken from participants' homes. (PDF)

**S2 Fig. Distribution of urinary DMA percent across cohorts.** a. DMA% distribution in three arsenic-exposed populations. The Health Effect of Arsenic Longitudinal Study (HEALS, in red), the New Hampshire Skin Cancer Study (NHSCS, in green), and the Strong Heart Study (SHS, in blue).
(PDF)

**S3 Fig. Non-imputed DMA% Conditional Association Results.** a. Results of a pre-imputation genetic association study of arsenic metabolism efficiency (DMA%) in the 10q24.32 region in three arsenic-exposed populations. P-values were generated with linear models adjusted for age and sex as well as kinship (HEALS) and population structure (SHS). The SNP with the strongest association is labeled in each panel. The top panel for each population shows the overall association results, the next shows p-values from models adjusted for the initial lead SNP, and the bottom panel shows the result of models adjusted for both previously identified variants. Three variants were identified in the Health Effect of Arsenic Longitudinal Study (HEALS), four in the Strong Heart Study (SHS), and one in New Hampshire Skin Cancer Study (NHSCS).
(PDF)

**S4 Fig. HEALS MMA% Conditional Association Results.** a. Results of a post-imputation genetic association study of MMA% in the 10q24.32 region in HEALS. P-values were generated with linear models adjusted for age, sex, and kinship. The top panel represents overall association results, the second shows p-values from models adjusted for the initial lead SNP, and the bottom panel shows the result of models adjusted for both previously identified variants. Two lead variants were identified in this analysis: chr10:102842818 and chr10:102888092. We further note that the primary signal likely captures the tertiary signal identified in the analysis of DMA%.
(PDF)

**S5 Fig. HEALS iAs% Conditional Association Results.** a. Results of a post-imputation genetic association study of iAS% in the 10q24.32 region in HEALS. P-values were generated with linear models adjusted for age, sex, and kinship. The top panel represents overall association results, the second shows p-values from models adjusted for the initial lead SNP, and the bottom panel shows the result of models adjusted for both previously identified variants. Three lead variants were identified in this analysis: chr10:103078084 (rs145537350), chr10:103089387 (rs12573221) and chr10:102853348. We further note that the primary and secondary signals are identical to two signals identified in the analysis of DMA%.
(PDF)

**S6 Fig. SHS MMA% Conditional Association Results.** a. Results of a post-imputation genetic association study of MMA% in the 10q24.32 region in SHS. P-values were generated with linear models adjusted for age, sex, and population structure. The top panel represents overall association results, the second shows p-values from models adjusted for the initial lead SNP, and the bottom panel shows the result of models adjusted for both previously identified variants. Two lead variants were identified in this analysis: chr10:103078084 (rs145537350) and

chr10:102842863 (rs4919688). We further note that the second signal is identical to the secondary signal identified in the analysis of DMA% and that the lead DMA% signal was the second most significant signal in the primary analysis and is in strong LD with the identified lead variant.
(PDF)

**S7 Fig. SHS iAs% Conditional Association Results.** a. Results of a post-imputation genetic association study of iAS% in the 10q24.32 region in SHS. P-values were generated with linear models adjusted for age, sex, and population structure. The top panel represents overall association results, the second shows p-values from models adjusted for the initial lead SNP, and the bottom panel shows the result of models adjusted for both previously identified variants. Two lead variants were identified in this analysis: chr10:103213304 and chr10:103015726.
(PDF)

**S8 Fig. Correlation of DMA% with arsenic exposure (based on drinking water arsenic concentration in HEALS (A) and NHSCS (B).** a. In both cohorts, we observe a negative correlation in which higher exposure is associated with lower DMA%.
(PDF)

**S9 Fig. Genomic annotations of HEALS confidence set 1.** a. Results of in silico-functional analysis reveals overlap with genomic annotations for candidate causal SNPs from HEALS confidence set 1, corresponding to the primary association signal (lead SNP rs145537350). Multiple SNPs in the confidence set overlap genomic features including candidate cis-regulatory elements and transcription factor binding sites. Highlighted panels show details of candidate causal SNPs overlapping these features.
(PDF)

**S10 Fig. Distribution of AS3MT expression across human tissues.** a. **A.** The distribution of AS3MT expression across human tissues reveals higher expression in the adrenal gland compared with all other available tissues. **B.** With the removal of the adrenal gland, we can see some variation in AS3MT expression across tissue types. Fig produced using the GTEx portal.
(PDF)

**S11 Fig. Co-localization between HEALS DMA% association signal (rs4919687) and *AS3MT* eQTLs in the Liver.** a. We detect *AS3MT* eQTLs in the liver and observe evidence of co-localization (Panel A), but this evidence was not consistent across all sets of priors analyzed. The low efficiency (low DMA%) allele at rs4919687 is associated with lower *AS3MT* expression in the liver, though this pattern is observed more strongly in other tissue types (Panel B).
(PDF)

**S1 Table. Minor Allele Frequency (MAF) comparison of lead signals across SHS centers.**
(DOCX)

**S2 Table. HEALS Confidence Sets (C.S.) from population-specific fine- mapping analysis.**
(DOCX)

**S3 Table. SHS Confidence Sets (C.S.) from population-specific fine- mapping analysis based on summary association statistics.**
(DOCX)

**S4 Table. SHS Confidence Sets (C.S.) from population-specific fine- mapping analysis based on primary sequencing data.**
(DOCX)

**S5 Table. Results of MANTRA meta-analysis of shared signal in HEALS and SHS.**
(DOCX)

**S6 Table. In-Silico Functional Examination of SHS Primary Sequencing Data Confidence Set 2.**
(DOCX)

**S7 Table. Co-localization of eQTLs for genes in the 10q24.32 region (excluding AS3MT) DMA% association signal (lead SNP rs4919687) identified in Bangladeshi individuals with posterior probability of colocalization > 80% (for p12 = 5x10-6).**
(DOCX)

**S8 Table. Co-localization (PP>80%) of mQTLs in the 10q24.32 region identified in GTEx tissues with the association signal for DMA% (lead SNP rs4919687) identified in a Bangladeshi population across a range of prior probabilities.**
(DOCX)

**S9 Table. Effect modification analysis of the effect of the interaction between DMA% associated variants and sex on DMA% in HEALS.**
(DOCX)

**S10 Table. Gene-Environment analysis of the effect of the interaction between DMA% associated variants and water arsenic concentration on DMA%.**
(DOCX)

**S11 Table. *AS3MT* eQTL Analysis Detailed Results.**
(XLSX)

**S12 Table. non-*AS3MT* gene eQTL Analysis Detailed Results.**
(XLSX)

**S13 Table. mQTL Analysis Detailed Results.**
(XLSX)

**S1 Data. Data Summary Statistics by Population.** a. HEALS Isolated Primary Association. b. HEALS Isolated Secondary Association. c. HEALS Tertiary Association. d. NHSCS Isolated Primary Association. e. SHS Isolated Primary Association. f. SHS Secondary Association.
(ZIP)

## Acknowledgments

We thank all the men and women who participated in the Health Effects of Arsenic Longitudinal Study, the New Hampshire Skin Cancer Study, and the Strong Heart Study and all the research staff who contributed to data collection and processing.

Web Resources

The Genotype-Tissue Expression (GTEx) Project: https://gtexportal.org/home/

FastQTL [50] (previously described by GTEx and downloaded from https://github.com/francois-a/fastqtl)

## Author Contributions

**Conceptualization:** Meytal Batya Chernoff, Brandon L. Pierce.

**Data curation:** Lin Tong, Muhammad G. Kibriya, Lei Huang.

**Formal analysis:** Meytal Batya Chernoff, Dayana Delgado, Lizeth I. Tamayo.

**Funding acquisition:** Habib Ahsan, Brandon L. Pierce.

**Investigation:** Dayana Delgado, Lizeth I. Tamayo, Farzana Jasmine, Muhammad G. Kibriya, Joseph Graziano.

**Methodology:** Meytal Batya Chernoff, Lin Chen, Meritxell Oliva.

**Project administration:** Brandon L. Pierce.

**Resources:** Lyle G. Best, Shelley Cole, Heather Nelson, Karin Haack, Jack Kent, Jason G. Umans, Joseph Graziano, Ana Navas-Acien, Margaret R. Karagas, Habib Ahsan.

**Software:** Lin Tong, Meritxell Oliva, Lei Huang.

**Supervision:** Brandon L. Pierce.

**Visualization:** Meytal Batya Chernoff.

**Writing – original draft:** Meytal Batya Chernoff.

**Writing – review & editing:** Lyle G. Best, Muhammad G. Kibriya, Ana Navas-Acien, Margaret R. Karagas, Brandon L. Pierce.

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
