## [Decision Letter · Decision Letter 0]

26 Jul 2022

Dear Dr Chernoff,

Thank you very much for submitting your Research Article entitled 'Sequencing-based fine-mapping and functional characterization of the 10q24.32 arsenic metabolism efficiency locus across multiple arsenic-exposed populations' to PLOS Genetics.

The manuscript was fully evaluated at the editorial level and by independent peer reviewers. The reviewers appreciated the attention to an important problem, but raised some substantial concerns about the current manuscript. Based on the reviews, we will not be able to accept this version of the manuscript, but we would be willing to review a much-revised version. We cannot, of course, promise publication at that time.

If you decide to revise the manuscript for further consideration at PLOS Genetics, please aim to resubmit within the next 60 days, unless it will take extra time to address the concerns of the reviewers, in which case we would appreciate an expected resubmission date by email to plosgenetics@plos.org.

[LINK]

We are sorry that we cannot be more positive about your manuscript at this stage. Please do not hesitate to contact us if you have any concerns or questions.

Yours sincerely,

Scott M. Williams

Section Editor

PLOS Genetics

Hua Tang

Section Editor

PLOS Genetics

Reviewer's Responses to Questions

**Comments to the Authors:**

Reviewer #1: Chernoff et al., have written a fascinating study aimed at understanding links between arsenic metabolism efficiency (AME) and genetic variation in the 10q24.32 region, near the arsenic methyltransferase ( AS3MT ) gene. They use fine mapping approaches to targeted sequencing data generated for arsenic exposed individuals from Bangladeshi, American Indian, and European American populations and add to the AME-associated SNPs, those that are associated with expression ad methylation through in silico analysis. There are many strengths of the study and a team of outstanding arsenic-focused scientists. I do have some suggestions to improve the manuscript.

Major suggested edits:

1. I would urge the authors to use caution with the use of terms such as “causal variants” and “functional characterization.” The causal variants were identified in relation to expression (not phenotype) and the functional characterization was carried out in silico and not using in vitro or in vivo techniques. It is strongly suggested to add “in silico” prior to the use of functional characterization throughout the manuscript including in the title to prevent from misleading the reader.

2. A description of other factors that may influence the differences in the finding of AME-associated SNPs/variants across studies is needed. The authors have done a good job listing some of the factors that could have influenced the finding of different AME-associated variants across populations. Please do expand this section to describe the following: (1) how differences in arsenic exposure levels might have influenced the results (e.g., very high exposure to low exposure), (2) how differences in sample sizes of the populations may have influenced the results (e.g., HEALS >1000, NH<250), and (3) how differences in subject recruitment and inclusion may have influenced the results (e.g. the NH study recruited those w/w/o skin cancer).

3. The authors examine AS3MT expression across tissues, but do not provide a differential analysis of the baseline expression in the tissues. This may impact the biological relevance of the findings. AS3MT has highly variable expression across tissues. The authors should provide the context for the baseline expression in these tissue and detail how these results impacted by potentially very low levels of AS3MT in various tissues.

4. The authors include the covariates of age and sex in their primary models. Please detail how these were selected for inclusion. That these factors are associated with AME is not in question. Rather, please provide support for these factors being associated with AS3MT genotype.

5. Related to the comment above, the cohorts vary significantly in BMI. How would the results change if obesity had been included in the models?

6. Please provide the beta estimate of rs191177668 in relation to AME in HEALS.

7. Please add the liver to Figure 6.

8. The eQTL analysis reads as if it is a biologically separate analysis from the identification of AME-associated SNPs. The authors should continue the information flowing from the analysis previously to add the estimates of association with gene expression for the identified AME-associated SNPs…(e.g., what was the estimate of rs191177668)

Minor suggested edits:

9. Additional citations are needed. Specifically, the authors state “In the cohorts with available measurements of arsenic in drinking water (HEALS and NHSCS) water arsenic exposure was inversely associated with DMA% (Fig S8), as reported previously”.. Please add the citation for where this was previously reported.

10. Needed text additions. In the abstract, it states that “There was one signal shared across all three examined populations.” But the exciting signal is not detailed. Realizing that this is complicated by the fact the one SNPs is shared between the SHS and NH cohorts but not directly in HEALS, it might make sense to list the common RS as well as the LD-associated RS in HEALS. Please add more information in the abstract stating which signal this is.

11. Also in the abstract, the authors state “causal variant across populations, and elucidate potential biological mechanisms” but do not state a mechanism for what? This would be improved by adding additional text to the end of this sentence

12. The authors state the arsenic metabolism is “believed” to occur primarily in the liver. Why do they use this qualifier?

13. Suggested wording modification from “is comprised of” to is composed of, or comprises.

14. The authors note “Prior studies have found that higher levels of MMA% and/or lower levels of DMA% are associated with increased risk for.” Here and throughout make sure to add the term urinary, as of course the proportions of these metabolites vary on tissue where liver, or target tissues versus urine would be very different.

15. No prior evidence suggests a role for BORCS7 in arsenic metabolism; …what about mirek’s paper

Reviewer #2: The authors investigate genetic determinants around the AS3MT locus on arsenic metabolism across three different study groups by combining seq data with arsenic metabolism phenotypes and bioinformatic analysis.

Major comments

It is very difficult to identify causal variants with low-to-moderate effect sizes across different populations when there are other factors that influence the phenotype of interest, namely the arsenic metabolism efficiency (AME). I have some comments on how the authors have handled this. According to table 1, there are individuals with a very high fraction of DMA in urine, i.e. >85%. It is very likely that among those individuals, some of the DMA does not come from inorganic arsenic, but from the metabolism of organic arsenic present in food stuff. Including those individuals in the SNP-AME analysis will blur the associations. The authors try to take this into account by adjusting for arsenobetaine, but there may still be residual confounding since the DMA and arsenobetaine are correlated in all 3 cohorts (no correlation coefficient is presented). I therefore suggest that you do a sensitivity analysis by excluding individuals with an arsenic metabolite pattern (>85%) that does not reflect the activity of AS3MT.

Both sex and smoking (smokers and men have a lower AME), and in some populations also the arsenic exposure level, affect the AME. It is not clear how smoking is handled in the association analyses. In table 1 it says “has smoked” - does this mean that all study participants were non- or ex-smokers? If not, did you do a sensitivity analysis excluding the current smokers? Further, did you stratify the analysis for men and women (you state that “the effects of our identified variants were similar in males and females”)? And did you also try to adjust for total sum of inorganic arsenic metabolites as you state that is influences the AME in some of the cohorts?

What is novel with the study should be better explained. The findings that AS3MT genotype influences DNA methylation and in turn gene expression of multiple genes in the AS3MT region has been known for a decade and this should be better acknowledged in the discussion. Further, it should be elaborated on how causal variant(s) can be identified further.

De Loma et al (2022) recently identified potentially functional SNPs in the AS3MT region linked to AME in South American populations. Are these polymorphisms in LD with the key ones identified here?

It is striking that AS3MT is expressed in so many tissues not primarily involved in the arsenic detoxification (e.g. the cardiovascular system and the brain). Moreover, you find multiple SNPs linked to %DMA as well as eQTL and mQTLs in different tissues, but few in the liver (this may reflect a database bias though). How do you explain this? Could it be that the SNPs are pleiotropic, both influencing AME and surrounding gene expression and in turn other pathways? And if so, how can you explain this co-regulation of genes with very different function, maintained by strong LD in the 10q32 region?

Minor edits

Introduction: Do not write AS3MT in italics when you refer to the protein.

**Have all data underlying the figures and results presented in the manuscript been provided?**

Reviewer #1: Yes

Reviewer #2: Yes

PLOS authors have the option to publish the peer review history of their article (what does this mean?). If published, this will include your full peer review and any attached files.

Reviewer #1: No

Reviewer #2: No

---

## [Decision Letter · Decision Letter 1]

20 Dec 2022

Dear Dr Chernoff,

We are pleased to inform you that your manuscript entitled "Sequencing-based fine-mapping and in silico functional characterization of the 10q24.32 arsenic metabolism efficiency locus across multiple arsenic-exposed populations" has been editorially accepted for publication in PLOS Genetics. Congratulations!

Yours sincerely,

Scott M. Williams

Section Editor

PLOS Genetics

Hua Tang

Section Editor

PLOS Genetics

Comments from the reviewers (if applicable):

Reviewer's Responses to Questions

**Comments to the Authors:**

Reviewer #2: The authors have responded to all my comments and I consider the manuscript acceptable for publication.

**Have all data underlying the figures and results presented in the manuscript been provided?**

Reviewer #2: Yes

PLOS authors have the option to publish the peer review history of their article (what does this mean?). If published, this will include your full peer review and any attached files.

Reviewer #2: No

**Data Deposition**

http://datadryad.org/submit?journalID=pgenetics&manu=PGENETICS-D-22-00626R1

**Press Queries**

---

## [Editor Report · Acceptance letter]

17 Jan 2023

PGENETICS-D-22-00626R1 

Sequencing-based fine-mapping and in silico functional characterization of the 10q24.32 arsenic metabolism efficiency locus across multiple arsenic-exposed populations 

Dear Dr Chernoff, 

We are pleased to inform you that your manuscript entitled "Sequencing-based fine-mapping and in silico functional characterization of the 10q24.32 arsenic metabolism efficiency locus across multiple arsenic-exposed populations" has been formally accepted for publication in PLOS Genetics! Your manuscript is now with our production department and you will be notified of the publication date in due course.

With kind regards,

Zsuzsanna Gémesi

PLOS Genetics

On behalf of:
